# BCG vaccination to reduce the impact of COVID-19 in healthcare workers: Protocol for a randomised controlled trial (BRACE trial)

Laure F Pittet ⓘ , Nicole L Messina ⓘ , Kaya Gardiner ⓘ , Francesca Orsini, Veronica Abruzzo, Samantha Bannister ⓘ , Marc Bonten, John L Campbell ⓘ , Julio Croda ⓘ , Margareth Dalcolmo ⓘ , Sonja Elia ⓘ , Susie Germano ⓘ , Casey Goodall, Amanda Gwee ⓘ , Tenaya Jamieson, Bruno Jardim ⓘ , Tobias R Kollmann ⓘ , Marcus Vinícius Guimarães Lacerda ⓘ , Katherine J Lee ⓘ , Donna Legge, Michaela Lucas ⓘ , David J Lynn ⓘ , Ellie McDonald ⓘ , Laurens Manning ⓘ , Craig F Munns ⓘ , Kirsten P Perrett ⓘ , Cristina Prat Aymerich ⓘ , Peter Richmond ⓘ , Frank Shann ⓘ , Eva Sudbury, Paola Villanueva ⓘ , Nicholas J Wood, Katherine Lieschke, Kanta Subbarao ⓘ , Andrew Davidson, Nigel Curtis ⓘ , the BRACE trial Consortium Group

**Correspondence to**
Professor Nigel Curtis;
nigel.curtis@rch.org.au

## ABSTRACT

**Introduction** BCG vaccination modulates immune responses to unrelated pathogens. This off-target effect could reduce the impact of emerging pathogens. As a readily available, inexpensive intervention that has a well-established safety profile, BCG is a good candidate for protecting healthcare workers (HCWs) and other vulnerable groups against COVID-19.

**Methods and analysis** This international multicentre phase III randomised controlled trial aims to determine if BCG vaccination reduces the incidence of symptomatic and severe COVID-19 at 6 months (co-primary outcomes) compared with no BCG vaccination. We plan to randomise 10 078 HCWs from Australia, The Netherlands, Spain, the UK and Brazil in a 1:1 ratio to BCG vaccination or no BCG (control group). The participants will be followed for 1 year with questionnaires and collection of blood samples. For any episode of illness, clinical details will be collected daily, and the participant will be tested for SARS-CoV-2 infection. The secondary objectives are to determine if BCG vaccination reduces the rate, incidence, and severity of any febrile or respiratory illness (including SARS-CoV-2), as well as work absenteeism. The safety of BCG vaccination in HCWs will also be evaluated. Immunological analyses will assess changes in the immune system following vaccination, and identify factors associated with susceptibility to or protection against SARS-CoV-2 and other infections.

**Ethics and dissemination** Ethical and governance approval will be obtained from participating sites. Results will be published in peer-reviewed open-access journals. The final cleaned and locked database will be deposited in a data sharing repository archiving system.

**Trial registration** ClinicalTrials.gov NCT04327206

## Strengths and limitations of this study

⇒ By including healthcare workers in five countries across three continents, this randomised controlled trial is large enough to assess the effect of BCG vaccination on the incidence of severe COVID-19, as well as symptomatic COVID-19 and other infections.

⇒ The trial includes robust clinical data collection on a daily basis during illness episodes to enable precise measurement of severity, as well as real-time tracking of missed opportunities for SARS-CoV-2 testing to trigger individualised reminders.

⇒ By including regular blood sampling, the trial will also provide information on the BCG-induced molecular and immunological changes associated with protection against off-target infections.

⇒ Limitations include the difficulty in blinding participants to group allocation due to the reaction and scar induced by BCG in most people, therefore requiring careful choice of objective outcomes as well as blinded assessment of measures where possible.

⇒ A further limitation is that aspects of the trial may require adjustment as the pandemic evolves and knowledge about COVID-19 expands.

## INTRODUCTION

In the twilight of 2019, the novel human pathogen SARS-CoV-2 emerged leading to the COVID-19 pandemic.[1] With no pre-existing immunity and in the absence of a vaccine, it was predicted that up to 60% of the global population could be infected.[2] As a result of their close contact with patients, healthcare workers (HCWs) are at particularly high risk.[3–5] By September 2020, over 7000 HCWs worldwide had succumbed to COVID-19.[6] This susceptibility is consistent with the

SARS epidemic in 2003, when HCWs comprised 21% of all cases.[4] HCW absenteeism or quarantine requirements due to COVID-19 impair healthcare delivery during the pandemic.

Preventive interventions to protect against emerging pathogens are needed, particularly for HCWs. The tuberculosis (TB) vaccine BCG has beneficial off-target effects that protect against unrelated infections.[7–15] These effects have been most extensively studied in high-mortality settings in Africa, where trials have shown a 38% reduction in all-cause neonatal mortality in infants vaccinated with BCG-Denmark compared with unvaccinated infants.[16 17] This protection, observed within days of vaccination, is proposed to be attributable to reduced deaths from infections other than those caused by *Mycobacterium tuberculosis*, particularly respiratory tract infections and sepsis.[7–9 16] The beneficial off-target effects of BCG are proposed to result from BCG-induced immunomodulation.[10 18–22] In adults, BCG vaccination increases innate immune responses to unrelated pathogens, an effect termed trained immunity,[23] that is sustained for at least a year after vaccination.[24]

In a human challenge model, prior vaccination with BCG-Denmark reduced viraemia by over 70% and improved antiviral immune responses to yellow fever virus vaccine,[10] a single-stranded, positive-sense RNA virus (as is SARS-CoV-2) compared with no BCG vaccination. In three randomised controlled trials in adults, BCG vaccination reduced the incidence of acute upper respiratory tract infections by 70%–80% compared with no BCG vaccination (although this was not necessarily the primary outcome of these trials).[11–13] Several studies have shown that BCG can also reduce symptoms in human papilloma virus and herpes simplex virus infections in adults.[14 15] Another study in adults showed that BCG-Bulgaria altered the clinical and immunological responses to malaria.[25] In animal models, numerous studies have shown that BCG protects against disease and mortality caused by a wide range of pathogens, including single-stranded, positive-sense RNA viruses.[15 26–28]

BCG vaccination potentially offers a readily available, safe, and low-cost way to reduce the incidence and/or severity of COVID-19, as well as other emerging pathogens that might arise in the future.[29 30] This would be of particular benefit among high-risk groups in whom the disease has the greatest impact, such as HCWs.

## STUDY AIMS
### Primary objective

To determine in HCWs if BCG vaccination compared with no BCG vaccination reduces the incidence of (1) Symptomatic and (2) Severe COVID-19 during the 6 months following randomisation.

### Secondary objectives

To determine in HCWs, during the 6 months and 12 months following randomisation, if BCG vaccination compared with no BCG vaccination: (1) Prolongs the time to first SARS-CoV-2-proven symptomatic infection; (2) Reduces the incidence and/or severity of febrile or respiratory illness, including COVID-19; (3) Reduces absence from work and (4) Is safe in HCWs (including revaccination). Exploratory objectives are to determine if BCG vaccination compared with no BCG vaccination: (1) Reduces oral herpes simplex virus reactivation in the subgroup of adults with recurrent cold sores; and (2) Changes immune function and whether these changes are associated with protection against non-tuberculous infectious diseases, including COVID-19. The study will also investigate factors that influence immune responses and infection risk (including COVID-19).

## METHODS AND ANALYSIS
### Trial design and setting

In this multicentre, phase III, randomised controlled trial, HCWs will be randomised in a 1:1 ratio to receive or not receive BCG vaccination. The protocol is available in the online supplemental material. Recruitment started 30 March 2020 and is divided into two stages. In stage 1 (March to April 2020), HCWs in Australia were randomised during the Australian influenza season in an open-label design to receive BCG and influenza vaccine, or influenza vaccine alone. In stage 2 (from May 2020), HCWs in five countries are randomised in a blinded fashion to receive BCG vaccine or placebo saline intradermal injection. Participants will be followed-up for 12 months. Data will be combined from both stages in a preplanned meta-analysis.

### Participants and eligibility criteria

Up to 10 078 HCWs in Australia, The Netherlands, Spain, the UK and Brazil will be recruited across both stages of the trial. Potential participants will receive information about the trial via email, healthcare facilities notice board, and/or website/social media. This will include a short description about the study, a link to a website with further information and contact details for questions. Potential participants will be able to evaluate their eligibility online, and access the site-specific participant information and consent form (see online supplemental material) prior to attending the clinic for enrolment. Eligibility will be ascertained by study staff during the baseline visit where participants will give informed consent. HCWs are eligible if working in healthcare settings during the COVID-19 pandemic or having face-to-face contact with patients. Stage 1 participants were required to receive influenza vaccination on the day of randomisation, regardless of group allocation. Exclusion criteria are: previous positive test for SARS-CoV-2, contraindication to BCG vaccine (eg, immunosuppression, pregnancy, serious underlying illness, history of active TB), previous adverse reaction to BCG vaccine (eg, significant local reaction

such as abscess or suppurative lymphadenitis), BCG vaccine administered within the last year, any other live-attenuated vaccine administered within the last month or indicated in the next month, any COVID-19-specific vaccine administered, or involvement in another COVID-19 prevention trial.

## Intervention

Participants randomised to BCG will receive a single dose of BCG-Denmark (AJ Vaccines, Copenhagen), 0.1 mL (corresponding to $2–8\times10^5$ colony forming units of *Mycobacterium bovis*, Danish strain 1331) as an intradermal injection over the region where the deltoid muscle inserts into the humerus, using a 1 mL syringe fitted with a short (10 mm) bevel needle (25G to 30G).

In stage 2, participants randomised to not receive BCG will be given a saline placebo intradermal injection using the same procedure described for BCG.

## Randomisation process

Randomisation to BCG or non-BCG groups will be done using a web-based randomisation procedure on the Research Electronic Data Capture platform (REDCap),[31] provided by an independent statistician. Randomisation will be in randomly permuted blocks of variable length (2, 4 or 6) stratified by stage of the study (stage 1 or 2), study site, age (<40 years; 40–59 years; ≥60 years) and presence of comorbidity (any of diabetes, chronic respiratory disease, cardiac condition, hypertension).

## Blinding

In stage 1, participants will be recruited in an open-label setting, meaning that only the trial statisticians will be blinded.

In stage 2, only those preparing and administering the intervention (BCG or placebo) will be unblinded; participants, investigators, statisticians and other trial staff involved in follow-up and data collection, will be blind to the randomisation group throughout the trial.

The code breaking procedure is available in the online supplemental material.

## Outcome

### Primary outcomes

Two primary outcomes have been chosen for this study: incidence of symptomatic COVID-19 and incidence of severe COVID-19 during the 6 months after randomisation. In light of the lack of knowledge about this new virus, we deemed it important to have sufficient power to detect the potential effect of BCG vaccine compared with no BCG (control group) on both of these outcomes. Our hypothesis is that BCG vaccine will reduce both the number of cases of symptomatic COVID-19 and the number of severe cases of COVID-19.

Symptomatic COVID-19 will be defined as an episode of illness with fever or at least one symptom of respiratory disease (including sore throat, cough and shortness of breath) plus a positive SARS-CoV-2 test (PCR, antigen or serology).

Severe COVID-19 will be defined as an episode of illness with a positive SARS-CoV-2 test plus at least one of the following as a consequence of COVID-19: (1) Death, (2) Hospitalisation or (3) Non-hospitalised severe disease, defined as being non-ambulant or unable to work for three consecutive days or more. Non-ambulant will be defined as being 'pretty much confined to bed (meaning finding it very difficult to do any normal daily activities)', and unable to work as 'not feeling physically well enough to go to work'.

### Secondary outcomes

Secondary outcomes will be assessed over the 6 months and 12 months following randomisation and are: any febrile or respiratory illnesses, duration of symptoms, number of days of absence from work, number of days confined to bed, complications (eg, pneumonia, need for oxygen therapy, admission to critical care, need for mechanical ventilation, outcome), and asymptomatic SARS-CoV-2 infection. Vaccine-related adverse reactions (frequency, severity and duration) will be compared between groups, and between participants who are BCG-naïve and those who are BCG-revaccinated.

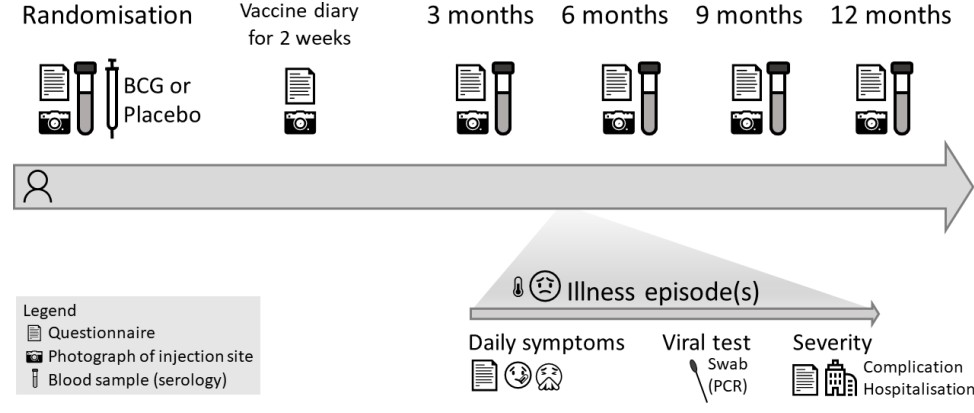

**Figure 1** Study flow chart.

### Exploratory outcomes

The impact of BCG vaccination on herpes simplex virus reactivation will be evaluated using the time to first recurrence, as well as the number, duration and severity of recurrences. The impact of vaccinations on the immune system will be evaluated using serology, immunoprofiling and cytokine responses.[21 22] The influence of host and external factors on the immune response and infection will also be evaluated.

### Data and sample collection

Participants will be followed-up for 12 months as illustrated in figure 1, using questionnaires and collection of blood samples. Additional information on severe disease will be obtained from hospital medical records.

### Questionnaires

Web-based questionnaires will be completed by participants or study staff at the time of recruitment,

randomisation, during the 2 weeks postvaccination (vaccine diary) and 3-monthly during follow-up, using the REDCap platform.[31] A summary of the data collected is provided in table 1. To verify eligibility, and for stratification prior to randomisation, the baseline questionnaire will collect data on demographics, workplace exposure and medical history. The vaccine diary will be used to document common reactions in the first 2 weeks after vaccination, with severity categorised using a toxicity grading scale.[32] At 3 months, 6 months, 9 months and 12 months after randomisation, follow-up questionnaires will be used to collect medical data outcome measures and potential modulating factors.

### Illness questionnaires

Participants will be asked weekly if they have been unwell since the last contact using a smartphone application designed for the trial (Trial Symptom Tracker, WeGuide)

**Table 1** Data collected from the questionnaires

| Participant questionnaires | Eligibility check | Baseline / randomisation | Vaccine diary* | 3-mly FU† | Episode FU |
|---|---|---|---|---|---|
| **Demographics, medical history** | | | | | |
| Inclusion and exclusion criteria | X | | | | |
| Age, sex, BMI, comorbidities (diabetes, cardiovascular disease, chronic respiratory disease, hypertension), alcohol, smoking | | X | | | |
| Medication: use of hydroxychloroquine, azithromycin, lopinavir-ritonavir, oseltamivir or antihypertensive drugs | | X | | X | |
| Vaccination: timing of administration of other vaccine(s) | | | | X | |
| *Mycobacterium* spp exposure: prior BCG vaccination, previous positive TST, latent TB, or stay in high TB prevalence country | | X | | X | |
| Cold sores recurrences and impact on quality of life | | X | | X | |
| **COVID-19 exposure** | | | | | |
| Workplace: profession, department, amount of direct patient contact, contact with COVID-19 cases | | X | | X | |
| Household: composition, COVID-19 exposure | | X | | X | |
| **BCG vaccination site** | | | | | |
| Photograph of BCG vaccination site | | X | X | X | |
| Vaccine site reaction (pain, redness, swelling, tenderness) | | | X | | |
| Enlarged lymph node | | | X | | |
| BCG scarring and complication | | | X | X | |
| **Illness episode** | | | | | |
| Presence of fever, cough, shortness of breath/difficulty breathing, sore throat, runny/blocked nose, fatigue, muscle or joint pain, headache, nausea or vomiting, diarrhoea, loss of smell or taste | | | | X | X |
| COVID-19 test result | | | | X | X |
| Days absent from work | | | | X | X |
| Days confined to bed | | | | | X |
| Medical consultation, ED presentation | | | | X | X |
| Hospital admission | | | | X | X |
| Complications | | | | X | X |

*Completed during the 2 weeks following vaccination.
†Questionnaire available in the online supplemental material.
BMI, body mass index; ED, emergency department; Episode, illness episode; FU, follow-up; mly, monthly; TB, tuberculosis; TST, tuberculin skin test.

and/or by contacting the participant by telephone, text message or email. With each episode of illness, the participant's symptoms will be recorded daily. At the end of the episode, a short survey will record COVID-19 test results, the severity of the episode, its management and its impact on ability to work.

### COVID-19 testing

When symptomatic, participants will be asked to undergo testing for SARS-CoV-2 infection with a validated test as required by their institution or local health authority. Participants who report fever or respiratory symptoms but have not been tested will be identified rapidly through the daily illness questionnaires and the participant will be called by the study staff to help arrange testing.

### Blood sampling

Blood will be collected at recruitment and 3 months, 6 months, 9 months and 12 months after randomisation for measurement of anti-SARS-CoV-2 antibodies. This will enable us to identify participants who were seropositive prior to randomisation and those who seroconvert during follow-up. Blood samples will also be used for the exploratory objectives related to vaccine-induced changes in the immune system and prediction of risk of COVID-19. Interferon $\gamma$ release assays will be included in sites with a high TB prevalence.

### Active tracking of missing data

Automated reports will identify in real time any missing data or missed opportunity for COVID-19 testing, enabling individualised reminders by email, text message and telephone call by the study staff.

### Sample size

The sample size for stage 2 of the study and the preplanned meta-analysis was chosen to provide adequate power for the two primary outcomes. Control of type I error will be managed by splitting the significance level between both outcomes and the preplanned interim analysis. The calculations are summarised in online supplemental table 1.

It is estimated that 55% of participants will have symptomatic COVID-19 in the control group by 6 months; with 1:1 randomisation, a sample size of 2016 will provide 95% power with two-tailed 0.005 significance level for the Pearson's $\chi^2$ test (with continuity correction) to detect an absolute difference of -10% (to 45%) between the BCG vaccine group and the control group.

For severe COVID-19 by 6 months, the study is powered to identify a risk ratio of 0.67 in the BCG compared with the control group. Assuming a 4% rate of severe COVID-19 by 6 months in the control group, a sample size of 6076 will provide 80% power with two-tailed 0.04 significance level for the Pearson's $\chi^2$ test to detect a risk ratio of 0.67, equivalent to an absolute difference of -1.3%. The remaining 0.005 was planned to be spent on

the interim analysis. Allowing for a 16% loss to follow-up, we plan to recruit 7244 HCWs in stage 2.

In the preplanned meta-analysis, there will be a total of 10078 participants (8062 allowing for 20% loss to follow-up). Assuming a 4% rate of severe COVID-19 by 6 months in the control group, a sample size of 8062 (4031 per group) will provide 90% power with two-tailed 0.04 significance level for the Pearson's $\chi^2$ test to detect a risk ratio of 0.67, equivalent to an absolute difference of -1.3%.

### Statistical analysis

The primary analysis will be an intention to treat analysis on all eligible participants randomised during stage 2 who did not have a SARS-CoV-2 positive test result at time of randomisation. The proportions of participants that meet each of the primary outcomes will be compared between the two groups using an absolute risk difference, estimated using a time-to-event analysis, where the survival curve for each combination of strata and randomised group will be calculated using a flexible parametric survival model (Royston-Parmar[33] model), presented with 95% CIs calculated with bootstrap standard errors.

Secondary analyses are planned using similar models, adjusting for the following covariates: sex, number and type of comorbidities, history of previous BCG vaccination. Subgroup analysis based on the same covariates are planned. The survival analysis strategy enables all participants with any follow-up data to be included in the analysis.

Secondary outcomes will be analysed and reported according to their nature (binary, continuous or categorical). Survival analysis will be used to analyse time to event outcomes with censoring of participants at their last follow-up.

A meta-analysis combining participants from both stages will be done, using the same analysis as described above, adjusted for the stage of the study.

Rate, severity, time to onset and duration of adverse reactions will be described using proportion, median and IQR, and expressed according to the number of participants answering the safety questions, presented by treatment group. The safety outcomes in the BCG group will be compared between participants who were revaccinated and those who were BCG-naïve using the $\chi^2$ test and the Mann-Whitney $U$ test.

Further exploratory analyses will evaluate the association between various factors and immune function, using both clinical and in vitro measures.

The full details of the interim and final analysis will be provided in the statistical analysis plan which will be finalised prior to unblinding and database lock.

### Interim analysis

There will be a single interim analysis of severe COVID-19 once 100 events have occurred.[34] The analysis will follow the same methodology as the primary analysis. The

stopping rule will be based on an alpha spending function, where the threshold to identify efficacy is based on the amount of data available at the time of the interim analysis.

## ETHICS AND DISSEMINATION
### Ethics
The trial will be run in accordance with the ethical principles of the Declaration of Helsinki, and ethical and governance approval sought for all participating sites. The primary Human Research Ethics Committee is the Royal Children's Hospital Melbourne (No. 62586), and the protocol was approved by all participating sites. An independent data safety monitoring committee (DSMC) will oversee trial conduct, safety and the interim analysis; the DSMC charter is available in the online supplemental material.

### Risk
The BCG vaccine has a well-established safety profile in healthy individuals. Infant BCG administration is near universal in many countries, and therefore many HCWs have previously received this vaccine. The risk of an earlier and accelerated local reaction is increased for HCWs who have previously had the vaccine (BCG revaccination), compared with those receiving it for the first time (BCG-naïve).[35–42] However, passive surveillance in countries recommending revaccination has not raised any safety concerns.[43 44] Also, rates of serious adverse events among BCG-revaccinated participants were not increased in large randomised controlled trials in Africa compared with those who had not previously received BCG.[11 45 46] Participants in the current study will not be tested for latent TB prior to inclusion as this does not predict the development of local skin reactions, abscesses or axillary lymphadenitis.[36] Of relevance for participants enrolled during stage 1, there is no additional risk for co-administration of influenza and BCG vaccines.[38 47]

While BCG has not been shown to cause fetal damage, the use of live-attenuated vaccines is contraindicated in pregnancy. Therefore, women of childbearing potential who think they could be pregnant or are planning to become pregnant within the next month are not eligible. Participants are asked to do a pregnancy test if they have any doubt, and encouraged to do so in the privacy of their homes. In some regions, completing a pregnancy test will be an eligibility requirement and test kits will be made available at recruitment.

There is a hypothetical risk that BCG-induced trained immunity could increase symptoms in those who contract SARS-CoV-2, leading to a higher incidence of symptomatic COVID-19 in the BCG-vaccinated compared with the non-vaccinated group. However, even if this occurs, BCG might still be associated with a lower risk of severe COVID-19 and hospitalisation as a result of a trained immune response reducing the viral load or clearing the infection faster. In a recent report, administration

of BCG at the time of hospital admission for COVID-19 did not raise any safety concerns.[48] In addition, there was no evidence of increased severity of COVID-19 among participants who participated in BCG trials prior to the pandemic.[49]

### Limitations
As a papule develops at the injection site around 2 weeks after BCG vaccination in most people, it is challenging to blind participants to their group allocation, even with a placebo. We have chosen objective primary outcomes to decrease the risk that awareness of allocation biases the trial results. Members of the research team following up the participants will be blinded to group allocation, as well as those doing the analysis (by the removal of all variables related to BCG from the data set) until data cleaning is complete and the statistical analysis plan has been signed by all investigators.

The BRACE trial is designed in two stages, from which data will be combined in a preplanned meta-analysis. Stage 2 was initiated when extending the recruitment outside Australia in April 2020. As it was spring in the northern hemisphere, the influenza vaccine could not be administered to the control group. In stage 2, participants are randomised to BCG vaccination or a saline placebo injection, with the placebo contributing to increasing the retention rate and lowering the risk of response bias.

The evolution of COVID-19 epidemiology and availability of COVID-19-specific vaccines are unpredictable, and we may not be able to detect any difference between the groups if the numbers of symptomatic and severe COVID-19 are too low. It is possible that HCWs may become less likely to contract SARS-CoV-2 as a result of improved preventive practices, reduced community transmission and/or following the availability of COVID-19 vaccines. In addition, once COVID-19-specific vaccines become available, interest in participating in the current trial might wane. We have therefore chosen diverse recruitment settings and included secondary outcome measures to evaluate the impact of BCG vaccination on other illnesses and infections (febrile and/or respiratory symptoms, herpes simplex virus reactivation) and on the immune system overall (immunological studies).

### Perspective
This trial is designed to determine whether BCG vaccination can reduce the incidence and/or severity of illness caused by SARS-CoV-2 infection. It also aims to provide data on the ability of BCG vaccination to reduce the overall rate and/or severity of febrile and respiratory illnesses in adults. This is particularly important for viral outbreaks that coincide with the winter influenza season, and could help to reduce the overall strain on the healthcare system. If the hypothesis of a beneficial effect of BCG vaccination is correct, then this vaccine could be implemented as an early intervention to protect HCWs and

other high-risk groups in future novel respiratory virus outbreaks.

## Dissemination

The trial protocol is registered at ClinicalTrials.gov (NCT04327206) and is available in the online supplemental material. Dissemination of the findings is planned, regardless of the results, through the WHO, in peer-reviewed journals and at scientific conferences. Once the database is cleaned and locked, it will be deposited in a data sharing repository archiving system. Access to the data will follow the rules of the repository system.

## Public involvement statement

The trial participants comprise only HCWs. The BRACE trial investigators include a number of HCWs and therefore representatives of the target population were heavily involved in the design, management and conduct of the trial. Most of the trial steering committee and the data safety and monitoring board (DSMB) members are also HCWs. The results of the trial will be disseminated to the trial participants and the HCW community.

**Correction notice** This article has been corrected since it first published. Members of 'the BRACE trial Consortium Group' have been updated.

**Twitter** Laure F Pittet @PittetLaure, Nicole L Messina @immunity_lab, Samantha Bannister @samibannister, Marc Bonten @MarcBonten, John L Campbell @profjcampbell, Julio Croda @JolioCroda, Susie Germano @GermanoSusie, Amanda Gwee @GweeAmanda, Bruno Jardim @BrunoJardim89, Katherine J Lee @Katherine_J_Lee, David J Lynn @DavidJohnLynn, Ellie McDonald @EllieMcDonald, Kirsten P Perrett @PerrettKirsten, Cristina Prat Aymerich @crisprat2010, Peter Richmond @PeterCRichmond and Nigel Curtis @nigeltwitt

**Acknowledgements** The authors thank the large number of people involved in establishing the BRACE Trial (see BRACE Trial Consortium Group list in the supplementary material), as well as Kathryn North, Ann Ginsberg and the Orygen team.

**Collaborators** The BRACE trial Consortium Group: Carolinne Abreu; Veronica Abruzzo; Lynne Addlem; Sophie Agius; Adelita Agripina Refosco Barbosa; Ahmed Alamrousi; Ayla Alcoforado da Silva dos Santos; Yasmeen Al-Hindawi; Samyra Almeida Da Silveira; Lais Alves da Cruz; Jeremy Anderson; Christina Anthony; Andrea Antonia Souza de Almeida dos Reis Pereira; Francisco Arnaiz de las Revillas Almajano; Annabelle Arnold; Beth Arrowsmith; Kristy Azzopardi; Cristina Badia Marti; Twinkle Bahaduri; Samantha Bannister; Sarah Barney; Lydia Barrera; Anabel Barriocanal; Dayanne Barros; Simone Barry; Adam Bartlett; Lilian Batista Silva Muranaka; Therese Baulman; Morgan Bealing; Justin Beardsley; Ana Belen Martin Gutierrez; Jason Bell; Saoirse Benson; Vicki Bennett-Wood; Nikki Bergant; Fabiane Bianca Barbosa; Wouter Bijllaardt; Patricia Bimboese; Camila Bitencourt de Andrade; Stephen Blake; Kitty Blauwendraat; Wim Boersma; Pilar Bohedo Garcia; Rhian Bonnici; Marc Bonten; Anne Boon; Anna Bourke; Kirsty Bowes; Larissa Brasil; Clare Brophy; Rochelle Botten; Sandy Buchanan; Jess Bucholc; Alison Burns; Emma Burrell; Natalia Bustos; Bridie Byrne; Anthony Byrne; Esther Calbo; Jorge Calvo Montes; Beatriz Camesella; John Campbell; Atsegiñe Cangas; John Carlin; Maria Carmen Roque; Roberta Carolina Pereira Diogo; Estela Carvalho; Irma Casas; Erika Castro; Ramon Castro; Helen Catterick; Rodrigo Cezar Dutra Escobar; Joyce Chan; Jo Cheah; Tee Yee Chern; Thilakavathi Chengodu; Marianna Ciaverella; Sharon Clark; Marie-Alix Clement Espindola; Annie Cobbledick; Clinton Colaco; Simone Collopy; Patricia Comella; Mary Corbett; Gabriela Correa E Castro; Erlane Costa; Raquel Coya; Nigel Crawford; Julio Croda; Alda Cruz; Nigel Curtis; Jac Cushnahan; Anna Czajko; Renato da Costa Silva; Bouchra Daitiri; Margareth Dalcolmo; Karen Dalton; Aiken Dao; Andrew Davidson; Phoebe Dawe; Diane Dawson; Miriam de Jesus Costa; Karina De La Cruz; Almudena de la Serna; Fabiani de Morais Batista; Adriely de Oliveira; Rocio del Alba Rey Morillo; Maria Desylva; Helga Dijkstra; Rachel Dixon; Maria Dolores del Toro Lopez; Jose Dominguez; Catriona Doran; Angel Dominguez Castellano; Glauce Dos Santos; Joyce dos Santos Lencina; Débora dos Santos Silva; Mark Douglas; Ross Dunn; Andrew Dunn; Jemma Dunnill; Georgina Eden; Harriet Edmund; Nat Eiffler; Hannah Elborough; Sonja Elia; Olivia Elkington; Michelle England; Wellyngthon Espindola Ayala; Maria Esteve; Nick Evans; Sue Evans; Krist Ewe; María Carmen Fariñ Álvarez; Kieran Fahey; Jill Fairweather; Denise Faustman; Erica Fernandes Silva; Monique Fernandez; Galina Fidler; P.M.G. Filius; Adam Finn; Carolyn Finucane; Stephanie Firth; Emily Fletcher; Catherine Flynn; Lorraine Flynn; Liam Fouracre; Sarah Fowler; Thamires Freitas; Ana Carolina Furtado; Maria Gabriela Oliveira; Anna Gabriela Santos; Leandro Galdino Cavalcanti Gonçalves; Laura Galletta; Larissa Gama; Dinusha Gamage; Radhika Ganpat; Carlos García; Mariana Garcia Croda; Kaya Gardiner; Evangeline Gardiner; Grace Gell; Aline Gerhardt de Oliveira; Susie Germano; Michael Gibbons; Camille Gibson; Alison Gifford; Teresa Giménez Poderos; Ann Ginsberg; Jet Gisolf; Bojana Gladanac; Penny Glenn; Vanessa Godinho; Mayara Góes dos Santos; Josune Goikoetxea; Telma Goldenberg; Adriano Gomes; Susana Gonzalez Marcos; Claudia González Rico; Casey Goodall; Louise Goodchild; Victoria Gordon; Frances Greven; Ana Greyce Capella; Liddy Griffith; Christina Guo; David Gutierrez Campos; Manuel Gutierrez Cuadra; Amanda Gwee; Richard Hall; Lydia Hall; Kate Hamilton; Matthew Hannan; Houda Harbech; Alex Harding; Neil Harker; Robert Harrison; Robert Jan Hassing; Thaynara Haynara Souza da Rosa; Zaheerah Haywood; Christine Heath; Nadine Henare; Paulo Henrique Andrade; Susan Herrmann; Erin Hill; Sam Hilton; Danique Huijbens; Heidi Hutton; Jane James; Tenaya Jamieson; Axel Janssen; Bruno Jardim; Tyane Jardim; Lance Jarvis; Narelle Jenkins; Jane Jones; Jan Jones; Karen Jones; Leticia Jorge; Maria Jose Rios Vilegas; Sri Joshi; Rosemary Joyce; Joel Junior; Rama Kandasamy; Anushka Karunanayake; Hana Karuppasamy; Tom Keeble; Jennifer Kent; Paul Kloeg; Jan Kluytmans; Bridget Knight; Tobias Kollmann; Tony Korman; Ann Krastev; Meredith Krieg; Nathan La; Marcus Lacerda; Alicia Lacoma; Renier Lagunday; Debbie Lalich; Erin Latkovic; Irene Latorre; Paulo Leandro Garcia Meireles Junior; Katherine Lee; Donna Legge; Toos Lemmers; Titia Leurink; Katherine Lieschke; Kee Lim; Gemma Lockhart; Cíntia Lopes Bogéa; Karla Lopes dos Santos; Reyes Lopez Marques; Michaela Lucas; David Lynn; Miriam Lynn; Maria Luciana Silva De Freitas; Norine Ma; Sam Macalister; Cristiane Machado; Matheus Machado Ramos; Francesca Machingaifa; Ivan Maia; Bernardo Maia; Richard Malley; Laurens Manning; Sarah Manton; Jose Manuel Carrerero; Ana Maria Barriocanal; Cíntia Maria Lopes Alves; Rosa Maria Plácido Pereira; Bianca Maria Silva Menezes Arruda; Adriana Marins; Angela Markow; Helen Marshall; Christopher Martin; Katya Martinez Almeida; Wayne Mather; Megan Mathers; Fábio Mauricio Nogueira Gomes; Mariana Mayumi Tadokoro; Nadia Mazarakis; Kelry Mazurega; Sonia McAlister; Amy McAndrews; Ellie McDonald; Fiona McDonald; Rebecca McElroy; Mark McMillan; Brendan McMullan; Nick McPhate; Lee Mead; Andrea Meehan; Bob Meek; Rosangela Melo; Guillermo Mena; Daniella Mesquita; Nicole L Messina; Isabella Mezzetti; Hugo Miguel Ramos Vieira; Skye Miller; Kirsten Mitchell; Marcus Mitchell; Jesutofunmi Mojeed; Kitty Molenaar; Gemma Molina; Barbara Molina; Lisa Montgomery; Cecilia Moore; James Moore; Simone Moorlag; Thilanka Morawakage; Julie Moss; Will Moyle; Kim Mulholland; Craig Munns; Elizandra Nascimento; Nicolas Navarrette; Mihai Netea; Juliana Neves; Georgina Newman; Belle Ngien; Jill Nguyen; Khanh Nguyen; Fran Noonan; Wendy Norton; Melissa O'Donnell; Jess O'Bryan; Abby O'Connell; Sasha Odoi; Liz O'Donnell; Roberto Oliveira; Marilena Oliveira; Thais Oliveira; Ingrid Oliveira; Nadia Olivier; Ligia Olivio; Benjamin Ong; Jaslyn Ong; Joanne Ong; Jakob Onysk; Isabelle Ooi; Frances Oppedisano; Francesca Orsini; Belinda Ortika; Orygen Group; Arthur Otsuka; Kristen Overton; Rosie Owens; Rayssa Paes; Pamela Palasanthiran; Virginia Palomo Jiménez; Girlene Pandine; Kimberley Parkin; Alvaro Pascual Hernandez; Nienke Paternotte; David Paterson; Ana Paula Conceição de Souza; Lisa Pelayo; Casey Pell; Sille Pelser; Handerson Pereira; Gabrielle Pereira; Glady Perez; Cristina Perez; Tomás Perez Porcuna; Susan Perlen; Kirsten Perrett; Amandine Philippart De Floy; Sigrid Pitkin; Laure F Pittet; R.C. Pon; Ines Portillo Calderón; Jeffrey Post; Catherine Power; Christiane Prado; Endriaen Prajitno; Cristina Prat-Aymerich; Lieke Preijers; Marco Puga; Evelyn Queiroz; Lynne Quinn; Ashleigh Rak; Leticia Ramires Figueiredo; Encarnacion Ramirez de Arellano; Pedro Ramos; Karla Regina Warszawski de Oliveira; Jack Ren; Stephanie Reynolds; Shelley Rhodes; Claudinalva Ribeiro dos Santos; Chris Richards; Peter Richmond; Holly Richmond; Ana Rita Lopes Souza; Jorge Rocha; Teresa Rodrigues; Laleyska Rodrigues; Bebeto Rodrigues; Iara Rodrigues Fernandes; Jesús Rodríguez-Baño; Nienke Roescher; Sally Rogers; Anke Rol; Jannie Romme; Antoni Rosell; Maria Roser Font; Domenic Sacca; Sonia Sallent; Vanderson Sampaio; Nuria Sanchez; Blanca Sanchez; Daniel Santos; Tilza Santos; Ariandra Sartim; Amber Sastry; Alice Sawka; Nikki Schultz; Clare Seamark; David Seamark; Engelien Septer-Bijleveld; Raquel Serrano; Frank Shann; Ketaki Sharma; Margaret Shave; Lisa Shen; Kate Sidaway-Lee; Adrian Siles Baena; Rafaela Silva; Juliana Silva; Emanuelle Silva; Mariana Simão; Ronita Singh; Marilda Siqueira; Marciléia Soares D.Allão Chaves; Thijs Sondag; Enoshini Sooriyarachchi; Antonny Sousa; Leena Spry; Sarah Statton; Andrew Steer;

Dyenyffer Stéffany Leopoldina dos Santos; Katrina Sterling; Leah Steve; Luke Stevens; Natalie Stevens; Carolyn Stewart; Lisa Stiglmayer; Lida Stooper; Josephine Studham; Kanta Subbarao; Eva Sudbury; Astrid Suiker; Lorrie Symons; Esther Taks; Niki Tan; Bruna Tayara Leopoldina Meireles; Menno te Riele; Jaap ten Oever; Guilherme Teodoro de Lima; Rob ter Heine; Jhenyfer Thalyta Campos Angelo; Helen Thomson; Ryan Toh; Alexandre Trindade; Harry Tripp; Enriqueta Tristán; Darren Troeman; Alexandra Truelove; Daniel Tsuha; Marlot Uffing; Fernando Val; Olga Valero; Ester Valls; Chantal van de Ven; Leo Van Den Heuvel; Sigrid van der Veen; Marije van der Waal; J.H. van Leusen; Linda van Mook; H. van Onzenoort; Marjoleine van Opdorp; Miranda van Rijen; Nicolette van Sluis; Adria Vasconcelos; Noelia Vega; Sunitha Velagapudi; Louise Vennells; Tamsin Venton; Harald Verheij; P.M. Verhoeven; Caroliny Veron Ramos; Paulo Victor Rocha da Silva; Sandra Vidal; Patricia Vieira; Matheus Vieira de Oliveira; Rosario Vigo Ortega; Paola Villanueva; Raquel Villar; Amanda Vlahos; Ushma Wadia; Mary Walker; Kate Wall; Rachael Wallace; Justin Waring; Ruth Warren; Adilia Warris; Emma Watts; Michelle Wearing-Smith; Daniel Webber-Rookes; Jamie Wedderburn; Ashleigh Wee-Hee; Steve Wesselingh; Jia Wei Teo; Bethany Whale; Phoebe Williams; Beatrijs Wolters; Nick Wood; Ivy Xie; Angela Younes; Angela Young; Felipe Zampieri Vieira Batista; Carmen Zhou; Vivian Zwart.

**Contributors** NC is the lead investigator and is responsible for study conception, design and funding acquisition. NC, AD, KS, LFP, NLM, KG, FS, KPP, AG, MB, JCa, JCr, MD, BJ, TRK, MVGL, ML, DJL, LM, CFM, CPA, PR, NJW, were involved in study design and outcome definition. SB, ES, PV, KPP, LFP, NC critically reviewed the literature for assessing the safety of BCG revaccination. KLi, LFP, NLM, FO, KLe, CG, EM, SB, PV, ES, KG, TJ, KPP, AD, NC prepared the ethics application and all other authors provided critical evaluation and revision. KG, VA, CG developed the recruitment methods. EM, CG, VA, LFP developed the questionnaire, smartphone application and trial database. SE, DL developed procedures for administration of the investigational product. NLM, SG designed the sample collection, processing methods and sample database. FO, KLe, LFP drafted the statistical analysis plan. LFP drafted the manuscript, coordinated manuscript preparation and revision. All authors provided critical evaluation and revision of the manuscript, and have approved the final version of this manuscript.

**Funding** This trial is supported by the Bill & Melinda Gates Foundation (INV-017302), the Minderoo Foundation (COV-001), Sarah and Lachlan Murdorch, the Royal Children's Hospital Foundation (2020-1263 BRACE Trial), Health Services Union NSW, the Peter Sowerby Foundation, the Ministry of Health Government of South Australia, the NAB Foundation, the Calvert-Jones Foundation, the Modara Pines Charitable Foundation, the UHG Foundation Pty Ltd, Epworth Healthcare and individual donors. NC is supported by the National Health and Medical Research Council (NHMRC) (Investigator Grant GNT1197117). LFP is supported by the Swiss National Science Foundation (Early Postdoc.Mobility grant number P2GEP3_178155). SB is supported by the University of Melbourne Research and Training Program scholarship and the Clifford Family scholarship. PV is supported by the Australian Government Research Training Program (RTP) Scholarship administered by the University of Melbourne and Murdoch Children's Research Institute PhD Top-Up Scholarship. The Melbourne WHO Collaborating Centre for Reference and Research on Influenza is supported by the Australian Government Department of Health (Health Protection Program Agreement # 4-4GFBE88).

**Competing interests** None declared.

**Patient consent for publication** Not applicable.

**Provenance and peer review** Not commissioned; externally peer reviewed.

**Data availability statement** Once the database is cleaned and locked, it will be deposited in a data sharing repository archiving system. Access to the data will follow the rules of the repository system.

**ORCID iDs**
Laure F Pittet http://orcid.org/0000-0002-2395-4574
Nicole L Messina http://orcid.org/0000-0001-8404-4462
Kaya Gardiner http://orcid.org/0000-0001-9796-4567
Samantha Bannister http://orcid.org/0000-0001-7507-3704
John L Campbell http://orcid.org/0000-0002-6752-3493
Julio Croda http://orcid.org/0000-0002-6665-6825
Margareth Dalcolmo http://orcid.org/0000-0002-6820-1082
Sonja Elia http://orcid.org/0000-0002-6368-189X
Susie Germano http://orcid.org/0000-0001-9924-7545
Amanda Gwee http://orcid.org/0000-0003-4016-8986
Bruno Jardim http://orcid.org/0000-0003-3465-7556
Tobias R Kollmann http://orcid.org/0000-0003-2403-9762
Marcus Vinícius Guimarães Lacerda http://orcid.org/0000-0003-3374-9985
Katherine J Lee http://orcid.org/0000-0002-8334-0291
Michaela Lucas http://orcid.org/0000-0001-8881-9990
David J Lynn http://orcid.org/0000-0003-4664-1404
Ellie McDonald http://orcid.org/0000-0003-2295-4641
Laurens Manning http://orcid.org/0000-0003-4334-5351
Craig F Munns http://orcid.org/0000-0001-5898-5808
Kirsten P Perrett http://orcid.org/0000-0002-5683-996X
Cristina Prat Aymerich http://orcid.org/0000-0001-6974-9165
Peter Richmond http://orcid.org/0000-0001-7562-7228
Frank Shann http://orcid.org/0000-0002-9899-1804
Paola Villanueva http://orcid.org/0000-0002-6504-2979
Kanta Subbarao http://orcid.org/0000-0003-1713-3056
Nigel Curtis http://orcid.org/0000-0003-3446-4594

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
