## [Reviewer comments · BMJ Open]

ARTICLE DETAILS

TITLE (PROVISIONAL)	BCG vaccination to reduce the impact of COVID-19 in healthcare workers: protocol for a randomised controlled trial (BRACE trial)
AUTHORS	Pittet, Laure; Messina, Nicole; Gardiner, Kaya; Orsini, Francesca; Abruzzo, Veronica; Bannister, Samantha; Bonten, Marc; Campbell, John; Croda, Julio; Dalcolmo, Margareth; Elia, Sonja; Germano, Susie; Goodall, Casey; Gwee, Amanda; Jamieson, Tenaya; Jardim, Bruno; Kollmann, Tobias; Guimarães Lacerda, Marcus Vinícius; Lee, Katherine; Legge, Donna; Lucas, Michaela; Lynn, David; McDonald, E; Manning, Laurens; Munns, Craig; Perrett, Kirsten; Prat Aymerich, Cristina; Richmond, Peter; Shann, Frank; Sudbury, Eva; Villanueva, Paola; Wood, Nicholas; Lieschke, Katherine; Subbarao, Kanta; Davidson, Andrew; Curtis, Nigel

VERSION 1 – REVIEW

REVIEWER	Rosendahl, AM University of Southern Denmark Faculty of Health Sciences, Department of Clinical Research
REVIEW RETURNED	04-Jun-2021

GENERAL COMMENTS	A well written and very thorough description of the study.
--

REVIEWER	Giamarellos-Bourboulis, Evangelos National and Kapodistrian University of Athens, 4th Department of Internal Medicine
REVIEW RETURNED	29-Jun-2021

GENERAL COMMENTS	This is an interesting trial where investigators aim to the benefit of BCG vaccination in the reduction of incidence and of severity of Covid-19. The vaccinated population are healthcare workers (HCW) who are at high exposure to Covid-19. I have some major concerns for this submission.  • Why testing for anti-SARS-CoV-2 is not used to exclude seropositive people? • Why there is no exclusion for people who are tuberculin positive, with stage IV malignancies or under corticosteroid treatment? • The investigators need to come up with a plan on how they will interpret antibody titers. If they are higher among BCG vaccinated people will this be interpreted as asymptomatic infection or as training allowing for better memory in case of infection or as protection from severe infection? • Will information on BCG vaccination at childhood captured? Will there be some subgroup analysis for these people? • I find a bit problematic the power calculation for stage 1. The investigators state that the incidence of Covid-19 will be 55%
---

	among placebo-vaccinated HCWs and that this will drop by 10% in the BCG vaccinated group. Why do they need to power so low as 0.0002?  • How did the investigators adjust for probable training benefit coming from influenza vaccine? • The investigators need to discuss if their results may be seen as development of a protection tool for future pandemics. They need to discuss how they plan to inform regulatory authorities around the world for these results. • What can be the antibody responses for someone who is already vaccinated with one specific anti-COVID-19 vaccine? Is there any chance that BCG can be harmful for these people?
--	---

VERSION 1 – AUTHOR RESPONSE

Reviewer: 1

Dr. AM Rosendahl, University of Southern Denmark Faculty of Health Sciences

Comments to the Author:

A well written and very thorough description of the study.

Authors' reply: Thank you for this positive comment.

Reviewer: 2

Dr. Evangelos Giamarellos-Bourboulis, National and Kapodistrian University of Athens

Comments to the Author:

This is an interesting trial where investigators aim to the benefit of BCG vaccination in the reduction of incidence and of severity of Covid-19. The vaccinated population are healthcare workers (HCW) who are at high exposure to Covid-19. I have some major concerns for this submission.

- Why testing for anti-SARS-CoV-2 is not used to exclude seropositive people?

Authors' reply: Anti-SARS-CoV-2 antibodies will be measured at inclusion (page 7 lines 330-331) and a positive test is an exclusion criterion (page 5 line 228).

- Why there is no exclusion for people who are tuberculin positive, with stage IV malignancies or under corticosteroid treatment?

Authors' reply: Exclusion criteria include all the contraindications to BCG vaccination, including immunosuppression, serious underlying illness, or history of active TB (page 5 lines 227-229). A positive tuberculin skin test is not an exclusion criterion as it is not a contraindication to BCG: studies show that it does not predict the risk of an adverse reaction after BCGa (pages 9-10 lines 434-435), a finding that has been confirmed in our participants (manuscript under review).

a Lotte A et al. BCG complications. Estimates of the risks among vaccinated subjects and statistical analysis of their main characteristics. Adv Tuberc Res 1984;21:107-93. (reference 34 in our manuscript)

- The investigators need to come up with a plan on how they will interpret antibody titers. If they are higher among BCG vaccinated people will this be interpreted as asymptomatic infection or as training allowing for better memory in case of infection or as protection from severe infection?

Authors' reply: SARS-CoV-2 antibody titers will principally be used to identify seroconversion rather than quantitative differences attributable to BCG training.

• Will information on BCG vaccination at childhood captured? Will there be some subgroup analysis for these people?

Authors' reply: Information on previous BCG vaccination is captured (Table 1 page 13). Subgroup analyses are planned for both the safety outcomes (page 9 lines 401-403) and the efficacy outcomes (now added on page 9 page 387-388).

• I find a bit problematic the power calculation for stage 1. The investigators state that the incidence of Covid-19 will be 55% among placebo-vaccinated HCWs and that this will drop by 10% in the BCG vaccinated group. Why do they need to power so low as 0.0002?

Authors' reply: We believe Reviewer 2 is referring to the sample size calculation of co-primary outcome 1 (symptomatic COVID-19) and not of stage 1. For co-primary outcome 1, the power was set at 95% to detect an absolute difference of 10% between an incidence of symptomatic COVID-19 disease with 2-tailed 0.005 significance level, which requires sample size of 2016 (page 8 lines 352-357). We deemed it important to have sufficient power to detect the potential effect of BCG vaccine for both co-primary outcomes. As the sample size required for co-primary outcome 2 (severe COVID-19) is greater than the 2016 required for co-primary outcome 1, the power for the latter is actually 100%.

• How did the investigators adjust for probable training benefit coming from influenza vaccine?

Authors' reply: In stage 1, participants were recruited while receiving influenza vaccine but this was not the case in stage 2. Both stages will be analysed separately, enabling us to evaluate the interaction between the intervention and the co-administration of influenza vaccine. Results from both stages will be combined in a pre-planned meta-analysis (page 9 line 396-397).

• The investigators need to discuss if their results may be seen as development of a protection tool for future pandemics. They need to discuss how they plan to inform regulatory authorities around the world for these results.

Authors' reply: This is discussed in the "Perspective" section (page 11, line 488-490). Regulatory authorities will be informed through the WHO and the Gates Foundation, who have been supporting the trial and following its progress closely (page 11, line 494-495).

• What can be the antibody responses for someone who is already vaccinated with one specific anti-COVID-19 vaccine? Is there any chance that BCG can be harmful for these people?

Authors' reply: Previous receipt of a COVID-19-specific vaccine is one of the exclusion criteria (page 5 lines 232-233). We collect data on COVID-19-specific vaccines received during the trial and this will be taken into account when interpreting serology results. We have chosen serological assays (including measuring anti-nucleocapsid antibody) that will enable us to differentiate COVID-19-specific vaccine responses from recovered COVID-19. Data on BCG vaccination's effect on COVID-19-specific vaccine responses and safety is also being collected and is included in our analysis plan.

VERSION 2 – REVIEW

REVIEWER	Giamarellos-Bourboulis, Evangelos National and Kapodistrian University of Athens, 4th Department of Internal Medicine
REVIEW RETURNED	29-Jul-2021
GENERAL COMMENTS	The authors have fully satisfied my concerns.